# Bias Aware Probabilistic Boolean Matrix Factorization

**Changlin Wan**[1,2]     **Pengtao Dang**[1]     **Tong Zhao**[3]     **Yong Zang**[1]     **Chi Zhang**[1]     **Sha Cao**[1]

[1]Indiana University, Indianapolis, Indiana, United States
[2]Purdue University, West Lafayette, Indiana, United States
[3]Amazon, Seattle, Washington, United States

## Abstract

Boolean matrix factorization (BMF) is a combinatorial problem arising from a wide range of applications including recommendation system, collaborative filtering, and dimensionality reduction. Currently, the noise model of existing BMF methods is often assumed to be homoscedastic; however, in real world data scenarios, the deviations of observed data from their true values are almost surely diverse due to stochastic noises, making each data point not equally suitable for fitting a model. In this case, it is not ideal to treat all data points as equally distributed. Motivated by such observations, we introduce a probabilistic BMF model that recognizes the object- and feature-wise bias distribution respectively, called bias aware BMF (BABF). To the best of our knowledge, BABF is the first approach for Boolean decomposition with consideration of the feature-wise and object-wise bias in binary data. We conducted experiments on datasets with different levels of background noise, bias level, and sizes of the signal patterns, to test the effectiveness of our method in various scenarios. We demonstrated that our model outperforms the state-of-the-art factorization methods in both accuracy and efficiency in recovering the original datasets, and the inferred bias level is highly significantly correlated with true existing bias in both simulated and real world datasets.

## 1 INTRODUCTION

Boolean matrix is one type of data representation with binary entries that originates from a wide range of applications including recommendation system, network analysis, collaborative filtering, and biological gene expression [Miettinen and Neumann, 2020, Balasubramaniam et al., 2018, Ko-

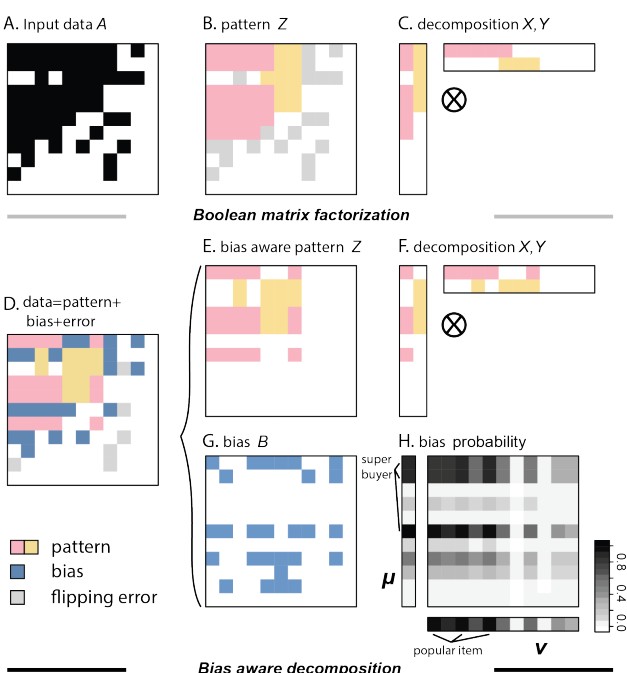

Figure 1: BMF with homoscedastic noise model (A-C) and bias aware BMF with column- and row-specific bias (D-H). H illustrates a biased data case in purchase history data.

cayusufoglu et al., 2018, Zhao et al., 2020, Liang et al., 2020]. The goal of **Boolean matrix factorization(BMF)** is to discover hidden patterns from binary data, where it finds a pair of low-rank binary matrices ($X \in \{0,1\}^{m \times k}$, $Y \in \{0,1\}^{k \times n}$) (Figure 1A,B,C), whose Boolean product approximates the original input matrix ($A \in \{0,1\}^{m \times n}$), i.e.,

$$A \sim X \otimes Y, \quad A_{ij} \sim \vee_{l=1}^{k} X_{il} \wedge Y_{lj}.$$

It has been known that under linear algebra, the input binary matrix $A$ can be of high rank, owing to the spike columns or rows, which prevents the application of established methods like SVD and PCA Wall et al. [2003]; while by applying BMF, in the most optimal case, one can reduce the rank of

*Accepted for the 38th Conference on Uncertainty in Artificial Intelligence* (UAI 2022).

the original matrix to its log level Monson et al. [1995]. Such low-rank decomposition can capture the local **dependency** between subsets of objects (row of $A$) and subsets of features (column of $A$). Specifically, in each rank-1 submatrix resulted from the decomposition into $X, Y$, i.e., $X_{:l} \otimes Y_{l:}$, it indicates a group of objects (i.e., nonzero entries in $X_{:l}$) sharing the same behavior on a set of features (i.e., nonzero entries in $Y_{l:}$). Here we denote the overall pattern matrix as $Z := X \otimes Y$. For the background error distribution, existing BMF methods tend to assume homoscedastic error distribution, or a universal flipping error with a flipping rate of $p_f = p(A_{ij} = 0 | Z_{ij} = 1) = p(A_{ij} = 1 | Z_{ij} = 0)$. In other words, the objective of BMF is to find the a decomposition of $A$ such that

$$A = (Z + E) \, mod \, 2 \, ; \, s.t. Z = X \otimes Y, p(E_{ij} = 1) = p_f$$

where $Z, E$ minimize a certain cost function $\tau(Z, A) = |E| = |A \ominus (X \otimes Y)|$ (Figure 1A,B,C). Here, $mod2$ represents the modulo operation with a quotient of 2, and $|\cdot|$ represents a certain norm measure defined by the cost function $\tau(\cdot)$.

Unfortunately, the assumption of homoscedastic error distribution is often violated when applied to complex real-world data, where the individual objects or features may have its specific bias pattern that result in **heteroscedastic error distribution**. Existing BMF methods fail to account for such object- or feature-specific bias, which could severely impact our ability to identify the true pattern $Z$, as the error matrix $E$ may display row- or column-specific bias [Wan et al., 2020a]. Take the online transaction records data as an example. The observed transaction records data from customers (row) and items (column) are constituted by three components: pattern, bias and flipping error (Figure 1D), meaning that aside from stochastic error, to determine whether or not a costumer would purchase a certain item, one should not only look at the purchase pattern that he/she belongs to (Figure 1E), but also his/her innate personal purchase preferences and the popularity of the item (Figure 1G). For example, a super-buyer, or someone with impulsive buying habits, is very likely to make a purchase regardless of the properties of the items; while a super-item, or a popular item, is also likely to be purchased by users with different characteristics (Figure 1B,H).

To mend the gap in binary data analysis, we propose **BABF** (**B**ias **A**ware **B**oolean matrix **F**actorization), the first tool to derive the latent binary pattern ($Z$), in the presence of individual row-wise and column-wise bias (Figure 1D-H), denoted as two real-valued probability vectors $\boldsymbol{\mu}, \boldsymbol{\nu}$, with $\boldsymbol{\mu}_i \in [0, 1] \, \forall i \in \{1, ..., m\}$ and $\boldsymbol{\nu}_j \in [0, 1] \, \forall j \in \{1, ..., n\}$. These two vectors represent processes that are object- and feature-specific, and are independent from the pattern generation process, or the homoscedastic background error. In other words, they capture the individual bias generation process that can't be captured by the existing model.

In this work, our contribution is three-fold:

- BABF is the first method that considers a heteroscedastic error model resulted from object- and feature-specific bias, which is more suitable for modeling real world data.
- BABF is a highly efficient algorithm in capturing the low rank structures in binary matrix in the presence of individual bias, and showed robust performance in deriving the true patterns across different data scenarios.
- As a byproduct of pattern discovery, BABF-derived individual bias patterns are highly consistent with the true bias pattern in simulated data and reasonable in real world data, which may lead to practical interpretations depending on different application scenarios.

## 2 PROBLEM FORMULATION

In this section, we formally address our objective to derive the latent Boolean patterns while considering the individual row- and column-wise bias in a probabilistic framework. We first introduce the notations used across this paper, then report the existing probabilistic BMF framework in Ravanbakhsh et al. [2016], Rukat et al. [2017], and then our bias-aware BMF model, BABF[1].

### 2.1 NOTATION

Matrix, vector and scalar values are denoted by uppercase ($A$), bold lowercase (**a**) and lowercase ($a$) characters, respectively. The upper-script represents the dimension of the object (e.g. $A^{m \times n}$), while the lower-script indicates the element indices (e.g. $i$-th row: $A_{i:}$, $j$-th column: $A_{:j}$, and $ij$-th element: $A_{ij}$). $|\cdot|$ represents a certain type of norm measure. Under Boolean arithmetic, the *and*, *or*, and *not* operations are denoted by $\wedge, \vee$, and $\neg$. Subsequently, the Boolean element-wise sum and subtraction are defined as $X \oplus Y = X \vee Y$ and $X \ominus Y = (\neg X \vee Y) \wedge (X \vee \neg Y)$. The Boolean matrix product is defined as $Z = X \otimes Y$, where $Z_{ij} = \vee_{l=1}^k X_{ik} \wedge Y_{lj}$.

### 2.2 EXISTING HOMOSCEDASTIC BMF MODEL

Following Ravanbakhsh et al. [2016], Rukat et al. [2017], each observed entry in a matrix $A$, i.e. $A_{ij} \in \{0, 1\}$, is assumed to be generated from the latent pattern $Z_{ij}$ with a homoscedastic error model with universal flipping probability $p_f$, where the likelihood function is defined as

$$p(A_{ij} | Z_{ij}) = \begin{cases} 1 - p_f, \, if \, A_{ij} = Z_{ij} \\ p_f, \, if \, A_{ij} \neq Z_{ij} \end{cases}$$

[1]code could be accessed at https://github.com/clwan/BABF

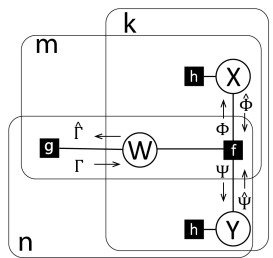

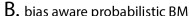

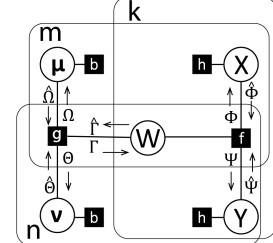

Figure 2: The factor graph representation of BMF and Bias-aware BMF. Noted, figure A is adopted from Ravanbakhsh et al. [2016]

$$p(A|Z) = \prod_{i,j} p(A_{ij}|Z_{ij})$$

As $Z = X \otimes Y$, individual Bernoulli prior is applied on every element of $X$ and $Y$, i.e.,

$$p(X) = \prod_{i,l} p(X_{ij}) \quad p(Y) = \prod_{l,j} Y_{jl}$$

Under this formulation, BMF is equivalent to a *Maximum A posterior (MAP)* inference problem of $X$ and $Y$ that maximizes the following overall likelihood function:

$$p(X, Y|A) \propto p(X)p(Y)p(Z|X,Y)p(A|Z)$$

Following Ravanbakhsh et al. [2016], we assume identical Bernoulli prior on $X, Y$, represented by factor $h$, e.g., $h(X_{il}) = log(p(X_{il}))$, $h(Y_{lj}) = log(p(Y_{lj}))$. Here, $p(Z|X,Y)$ encodes the hard constraint that ensures the equality of the Boolean product, i.e., $Z = X \otimes Y$. By introducing an auxiliary tensor $W \in \{0,1\}^{m \times n \times k}$, where $W_{ijl} = X_{il} \wedge Y_{lj}$, $Z_{ij} = \vee_{l=1}^{k} W_{ijl}$, this hard constraint is dispersed onto each element in $W$, and can be reformulated as an identity constraint as

$$p(W_{ijl}|X_{il}, Y_{lj}) = \mathcal{I}(W_{ijl} = X_{il} \wedge Y_{lj})$$

where for $\mathcal{I}$, we have $\mathcal{I}(true) = 1$ and $\mathcal{I}(false) = 0$. Obviously, if $W_{ijl} \neq X_{il} \wedge Y_{lj}$, the factor $f(W_{ijl}, X_{il}, Y_{lj}) = log(p(W_{ijl}|X_{il}, Y_{lj}))$ will be evaluated to be $-\infty$. Finally, factor $g(\{W_{ijl}\}, \forall l \in \{1, ..., k\}) = log(p(A_{ij}|Z_{ij}))$ assess the likelihood of observed variable $A_{ij}$ given the latent pattern $Z_{ij}$. Overall, we have the factor graph representation of the log-likelihood $p(X, Y|A)$ (Figure 2A, adopted from Ravanbakhsh et al. [2016]) as

$$log(p(X, Y|A)) = \sum_{ij} h(X_{ij}) + \sum_{lj} h(Y_{lj})$$
$$+ \sum_{ijl} f(W_{ijl}, X_{il}, Y_{lj}) + \sum_{ij} g(\{W_{ijl}\}_l)$$

Owing to the NP-hard complexity of BMF [Stockmeyer, 1975, Gillis and Vavasis, 2018], it is intractable to infer the

MAP of the log-likelihood. Alternatively, focusing on the marginal-MAP often yields good empirical success [Ravanbakhsh et al., 2016, Rukat et al., 2017], e.g.,

$$\arg\max_{X_{il}} log(p(X_{il}|A)) =$$
$$\arg\max_{X_{il}} \sum_{X_{i'l'} \setminus X_{il}, Y_{l'j'}} log(p(X_{i'l'}, Y_{l'j'}|A))$$

Max-sum belief propagation and Gibbs sampling have been reported to achieve good performance under such a strategy [Ravanbakhsh et al., 2016, Rukat et al., 2017].

## 2.3 PROPOSED BIAS AWARE BMF MODEL

The probabilistic BMF model presented above provides a good framework for us to account for the feature- and object-wise bias. Compared with the homoscedastic setting, the core advancement of our work is to consider the observed data as generated from a process that is more realistic: aside from stochastic error, or the homoscedastic error distribution as in [Ravanbakhsh et al., 2016, Rukat et al., 2017], we consider that the observed data is generated not only from the latent pattern $Z = X \otimes Y$, but also from independent object/feature behavior process governed by a bias matrix $B \in \{0,1\}^{m \times n}$, where $B$ is determined by a row- and column-wise bias vector $\boldsymbol{\mu}$ and $\boldsymbol{\nu}$ in such way that

$$p_{B_{ij}} = p(B_{ij} = 1) = \boldsymbol{\mu}_i \boldsymbol{\nu}_j$$

And the generation process of $A$ is hence

$$A = B \oplus ((Z + E) \, mod \, 2).$$

The new likelihood of each observations can be characterized in the following four scenarios:

$$p(A_{ij} = 1|Z_{ij} = 0) = 1 - (1 - p_f)(1 - \boldsymbol{\mu}_i \boldsymbol{\nu}_j)$$
$$p(A_{ij} = 0|Z_{ij} = 0) = (1 - p_f)(1 - \boldsymbol{\mu}_i \boldsymbol{\nu}_j)$$
$$p(A_{ij} = 1|Z_{ij} = 1) = 1 - p_f(1 - \boldsymbol{\mu}_i \boldsymbol{\nu}_j)$$
$$p(A_{ij} = 0|Z_{ij} = 1) = p_f(1 - \boldsymbol{\mu}_i \boldsymbol{\nu}_j)$$

The new posterior probability could then be written as

$$p(X, Y, \boldsymbol{\mu}, \boldsymbol{\nu}|A) =$$
$$p(X)p(Y)p(Z|X,Y)p(\boldsymbol{\mu})p(\boldsymbol{\nu})p(A|Z, \boldsymbol{\mu}, \boldsymbol{\nu})$$

Factor graph representation of the new posterior is shown in Figure 2B. Comparing to the existing probabilistic BMF model introduced in 2.2, the new factor graph involves the row- and column-wise bias vectors $\boldsymbol{\mu}, \boldsymbol{\nu}$. Given no prior knowledge of the two variables, we assume a uniform prior on $\boldsymbol{\mu}, \boldsymbol{\nu}$, thus factor $b(\boldsymbol{\mu}_i), b(\boldsymbol{\nu}_j)$ evaluate to 0 in the graph. And the likelihood factor $g$ is also related to $\boldsymbol{\mu}, \boldsymbol{\nu}$ in the new formulation. In the next section, we introduce BABF algorithm to derive the decomposition.

# 3 THE ALGORITHM OF BABF

While we assume $A$ to be generated from two sources, latent pattern $Z$ and Bias $B$, these two sources themselves can be considered as independent from each other. Such independence is also reflected on the factor graph (Figure 2B). Though the likelihood factor $g$ and the auxiliary variables $W$ are involved with both $\{X, Y\}$ and $\{\boldsymbol{\mu}, \boldsymbol{\nu}\}$, the direct message update of $\{X, Y\}$ and $\{\boldsymbol{\mu}, \boldsymbol{\nu}\}$ are independent with each other. Conveniently, $\{X, Y, W\}$ and $\{\boldsymbol{\mu}, \boldsymbol{\nu}, W\}$ can be considered as two individual systems to be treated separately.

---

**Algorithm 1:** BABF, Bias Aware BMF

---

**Inputs:** $A, k, p_X, p_Y, p_f, t_{all}, t_{MF}, t_{BI}$

**BABF**:

**while** $t \leq t_{all}$ *and not converged messages*

    $\boldsymbol{\mu}^{t+1}, \boldsymbol{\nu}^{t+1} \leftarrow$ Bias_infer$(A, X^t, Y^t, t_{BI})$

    $X^{t+1}, Y^{t+1} \leftarrow$

    prob_BMF$(A, k, p_X, p_Y, p_f, \boldsymbol{\mu}^{t+1}, \boldsymbol{\nu}^{t+1}, t_{MF})$

**end**

**Bias_infer**:

$Z := X \otimes Y$

**while** $t \leq t_{BI}$ *and* $error\_now < error\_all$

    $error\_all := error\_now$

    $\boldsymbol{\mu}_i^{t+1} := \frac{\sum_{j \in \{j_0 | Z_{ij_0} = 0\}} A_{ij} \boldsymbol{\nu}_j^t}{\sum_{j \in \{j_0 | Z_{ij_0} = 0\}} \boldsymbol{\nu}_j^t}, \forall i \in \{1, ..., m\}$

    $\boldsymbol{\nu}_j^{t+1} := \frac{\sum_{i \in \{i_0 | Z_{i_0 j} = 0\}} A_{ij} \boldsymbol{\mu}_i^t}{\sum_{i \in \{i_0 | Z_{i_0 j} = 0\}} \boldsymbol{\mu}_i^t}, \forall j \in \{1, ..., n\}$

    $error\_now :=$

    $\sum_{(i,j) \in \{(i_0, j_0) | Z_{i_0 j_0} = 0\}} (A_{ij} - \boldsymbol{\mu}_i \boldsymbol{\nu}_j)^2$

**end**

**prob_BMF**:

$p(A_{ij} | Z_{ij}) \leftarrow$ calculate based on $\boldsymbol{\mu}, \boldsymbol{\nu}$.

Initialize $\Psi_{ijl}^0, \hat{\Psi}_{ijl}^0, \Phi_{ijl}^0, \hat{\Phi}_{ijl}^0, \Gamma_{ijl}^0, \hat{\Gamma}_{ijl}^0, \forall i, j, l$

**while** $t \leq t_{MF}$ *and not converged messages*

    $\Phi_{ijl}^{t+1} := \max(\Gamma_{ijl}^t + \hat{\Psi}_{ijl}^t, 0) - \max(\Psi_{ijl}^t, 0)$

    $\Psi_{ijl}^{t+1} := \max(\Gamma_{ijl}^t + \hat{\Phi}_{ijl}^t, 0) - \max(\Phi_{ijl}^t, 0)$

    $\hat{\Phi}_{ijl}^{t+1} := log(\frac{1-p_f}{p_f}) + \sum_{j' \neq j} \Phi_{ij'l}^t$

    $\hat{\Psi}_{ijl}^{t+1} := log(\frac{1-p_f}{p_f}) + \sum_{i' \neq i} \Psi_{i'jl}^t$

    $\Gamma_{ijl}^{t+1} := \min(log(\frac{p(A_{ij}|1)}{p(A_{ij}|0)} +$

    $\sum_{l' \neq l} \max(\Gamma_{ijl'}^t)), \max(0, - \max_{l' \neq l} \hat{\Gamma}_{ijl'}^t))$

    $\hat{\Gamma}_{ijl}^{t+1} := \min(\hat{\Phi}_{ijl}^t + \hat{\Psi}_{ijl}^t, \hat{\Psi}_{ijl}^k, \hat{\Phi}_{ijl}^t)$

**end**

$X_{il} = \begin{cases} 1, & \text{if } log(\frac{1-p_f}{p_f}) + \sum_i \Phi_{ijl}^t > 0 \\ 0, & otherwise. \end{cases}$

$Y_{lj} = \begin{cases} 1, & \text{if } log(\frac{1-p_f}{p_f}) + \sum_i \Psi_{ijl}^t > 0 \\ 0, & otherwise. \end{cases}$

**Outputs:** $X, Y, \boldsymbol{\mu}, \boldsymbol{\nu}$

---

Under this formulation, fitting pattern $\{X, Y\}$ while given $B$ is an NP-hard problem as it can be regarded as traditional BMF without the influence of $B$, i.e.,

$$A \cdot (\neg B) = ((Z + E) \bmod 2) \cdot (\neg B).$$

Or probabilistically, while given $B$, this problem could be reduced to weighted graph maximum cut, which is also NP hard Stockmeyer [1975], Gillis and Vavasis [2018]. Overall, we can claim bias-aware BMF is at least as hard as traditional BMF. Therefore, it is also an NP hard problem. To solve this problem, we still turn to find the marginal-MAP, which corresponds to optimally estimating individual variables, while the other variables are marginalized.

Here we introduce BABF in algorithm 1. BABF has two core components, prob_BMF and Bias_infer, corresponding to the derivations of $\{X, Y\}$ and $\{\boldsymbol{\mu}, \boldsymbol{\nu}\}$. Other than the input data $A$, BABF takes the pattern number parameters $k$, the Bernoulli prior of $X, Y$, filling error $p_f$ and the maximum iterations for the overall algorithm as well as core components $(t_{all}, t_{MF}, t_{BI})$ as input, and outputs the decomposition $X, Y$ and the bias vectors $\boldsymbol{\mu}, \boldsymbol{\nu}$.

## 3.1 PROB_BMF

When fixing bias vectors $\boldsymbol{\mu}, \boldsymbol{\nu}$, the only differences between bias aware BMF and the existing BMF model introduced in 2.2 is that each likelihood factor $g_{ij}$ would evaluate to different probability assignments by referencing $\boldsymbol{\mu}_i, \boldsymbol{\nu}_j$. Following Ravanbakhsh et al. [2016], we utilize the max-sum belief propagation (BP) strategy to approximate the overall likelihood in prob_BMF. Correspondingly, the message $\Gamma_{ijl}$ that propagates the likelihood information to auxiliary variable $W$ would be different from Ravanbakhsh et al. [2016] with individualized probabilities. We introduced detailed derivations of the message passing process in the Bias aware factor graph (Figure 2B) in the Appendix.

## 3.2 BIAS_INFER

The inference of the marginal-MAP of $\boldsymbol{\mu}, \boldsymbol{\nu}$ is a non-trivial task even with accurate pattern information $Z$, as for any bias variable $\boldsymbol{\mu}_i$, any observation related to this variable is related to a different $\boldsymbol{\nu}_i$, and vice versa. To circumvent this computational challenge, we adopted two modifications. 1) We only consider the observations that are not covered by pattern $Z$ for bias inference. We argue the pattern related observations have marginal contribution to the bias inference and could be omitted. 2) Instead of deriving exact MAP, we treat this as an optimization problem, where we could utilize conventional loss functions to achieve the same objective that optimize the difference between $\boldsymbol{\mu}, \boldsymbol{\nu}$ and background information. Inspired by Wan et al. [2020a], we apply a modified mean square loss. Take $\boldsymbol{\mu}_i$ as an example, the loss

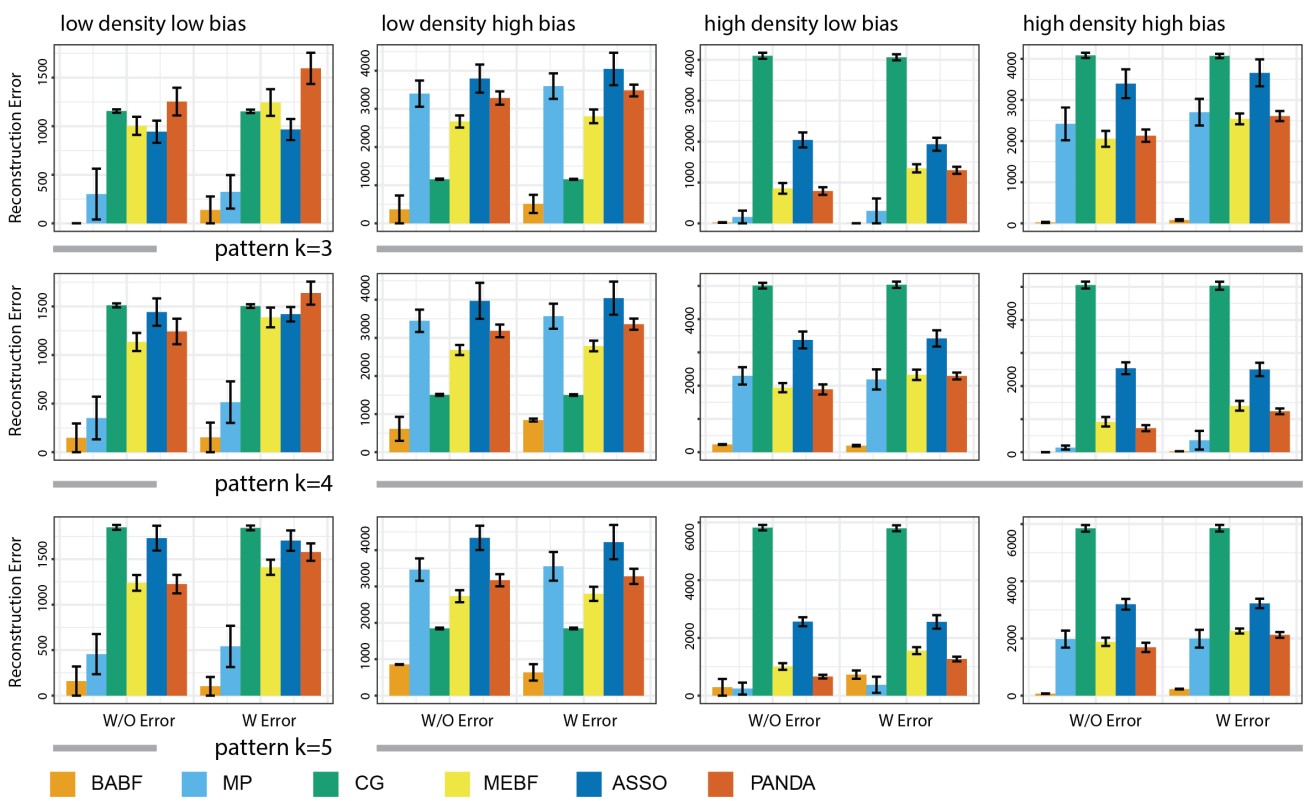

Figure 3: Performance comparison on simulated data

function takes the form of

$$\Omega_i = \sum_{j \in \{j_0 | Z_{ij_0} = 0\}} \boldsymbol{\nu}_j^t (A_{ij} - \boldsymbol{\mu}_i^t)^2$$

The most important benefit of this modified loss is that it ensures each probability $\boldsymbol{\mu}_i$ would be from the interval [0,1], and it still considers the impact of $\boldsymbol{\nu}_j$ on each observation $A_{ij}$. Moreover, it is with high computational feasibility as the updated $\boldsymbol{\mu}_i^{t+1}$ could be easily derived as $\boldsymbol{\mu}_i^{t+1} := \frac{\sum_{j \in \{j_0 | z_{ij_0} = 0\}} A_{ij} \boldsymbol{\nu}_j^t}{\sum_{j \in \{j_0 | z_{ij_0} = 0\}} \boldsymbol{\nu}_j^t}$. And similarly, $\boldsymbol{\nu}_j^{t+1} := \frac{\sum_{i \in \{i_0 | z_{i_0 j} = 0\}} A_{ij} \boldsymbol{\mu}_i^t}{\sum_{i \in \{i_0 | z_{i_0 j} = 0\}} \boldsymbol{\mu}_i^t}$. Here, we implement this strategy in Bias_infer. Empirically, it is robust for the bias inference across different scenarios, which we will introduce in detail in the Experiments section.

### 3.3 COMPLEXITY ANALYSIS

The computational cost of BABF depends on the core modules. For each iteration, prob_BMF will visit all variables in $\{X, Y, W\}$, and the calculation of the message update is at constant cost. Hence, the cost of prob_BMF is bounded by the size of latent variables, i.e. $O(mnk)$. The consideration of mean square loss enables high computational feasibility to update the bias, therefore, in each iteration of Bias_infer

the computation is linear with data size, i.e., $O(mn)$. Overall, the computational cost of each iteration of BABF is $O(mnk)$.

## 4 EXPERIMENTS

We evaluate the performance of our bias aware model on both synthetic and real world datasets. We first introduce related methods for BMF and report the benchmark performance across different simulated data scenarios. We then highlight the practical use of BABF in our analysis of a movielens and gene expression data.

### 4.1 RELATED WORK

In addition to the probabilistic methods introduced above [Ravanbakhsh et al., 2016, Rukat et al., 2017], different heuristic methods have been developed to solve the BMF problem. Previously Wan et al. [2020a] systematically discussed the bias issue in BMF, but their focus is to explore the identifiability of the patterns in the presence of bias in the noise model. For the rest of the methods, none of them considered the heteroscesdastic issue of the error distribution. Among these methods, ASSO represents a series of work from Miettinen et al [Miettinen et al., 2008, Miettinen and Vreeken, 2011, Karaev et al., 2015, Tatti and Miettinen,

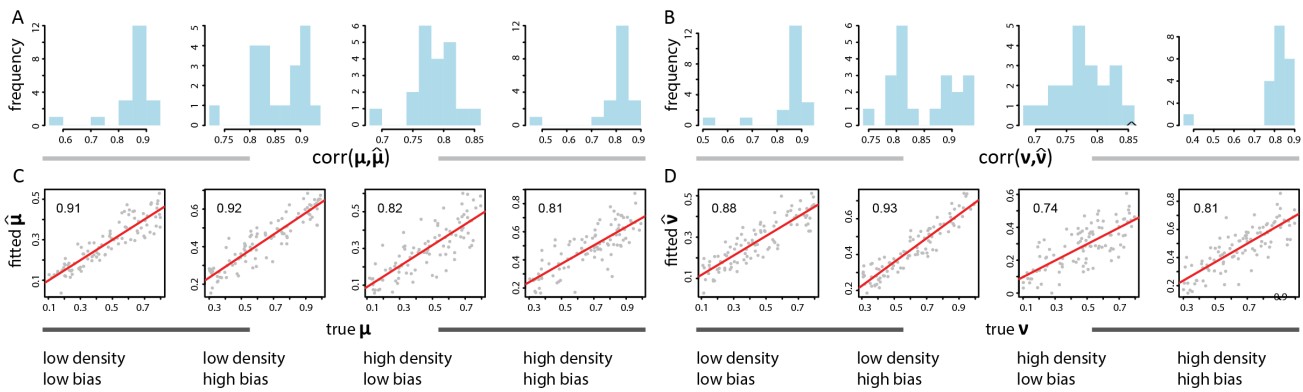

Figure 4: BABF inferred bias is highly correlated with ground truth bias

2019]. ASSO first generates a pool of column bases from row-wise correlation matrix, and iteratively searches for the best column and row bases following a pre-defined cost function. PANDA is another series of heuristic methods that embed the cost function in the search of top_$k$ core patterns [Lucchese et al., 2010, 2013]. Formal Concept Analysis also showed empirical success in BMF [Belohlavek and Trnecka, 2015, 2018, Belohlavek et al., 2019]. More recently, Wan et al. [2020b] proposed a fast algorithm by formulating submatrix pattern identification in a geometric perspective. Kovacs et al. [2020] formulates BMF as an integer program problem and utilizes column generation framework to search for the best solutions. Here, we benchmark the performance of BABF with MP [Ravanbakhsh et al., 2016], CG [Kovacs et al., 2020], MEBF [Wan et al., 2020b], ASSO [Miettinen et al., 2008] and PANDA [Lucchese et al., 2010] and believe that this set of methods represent the state-of-the-art performance of BMF in different perspectives.

## 4.2 BENCHMARK ON SIMULATED DATA

We simulate an observed binary matrix $A$ by the following model:
$$A = B \oplus ((Z + E) \bmod 2).$$

Here, $B, Z, E$ represent the column-/row-wise bias matrix, pattern matrix and error matrix respectively. Each entry in $B, E \in \{0, 1\}^{m \times n}$ is simulated to follow Bernoulli distribution with success probabilities $p(B_{ij}) \propto \boldsymbol{\mu}_i \boldsymbol{\nu}_j$ and $p(E_{ij}) = p_f$. The latent pattern matrix is generated by $Z = X \otimes Y$, where $X \in \{0, 1\}^{m \times k}, Y \in \{0, 1\}^{k \times n}$, and entries in $X, Y$ also follow Bernoulli distributions with success probabilities $p(X_{il}) = p_X$ and $p(Y_{lj}) = p_Y$. To comprehensively evaluate the methods, we generate varied data scenarios by considering different pattern numbers ($k \in \{3, 4, 5\}$), and flipping error ($p_f \in \{0, 0.05\}$). We also use different levels of $p_X, p_Y$ to simulate pattern matrices of different density levels, where low density has $p_X = p_Y = 0.2$ while high density has $p_X = p_Y = 0.4$. The bias level is controlled by $\boldsymbol{\mu}, \boldsymbol{\nu}$. In case of low bias,

we sample every $\boldsymbol{\mu}_i, \boldsymbol{\nu}_j$ uniformly from $[0.1, 0.8]$, which yields a overall bias level of $p_{\bar{B}_{ij}} \sim 0.2$. For the high bias case, $\boldsymbol{\mu}_i, \boldsymbol{\nu}_j$ is sampled from $[0.3, 0.9]$ that results an overall bias level of $p_{\bar{B}_{ij}} \sim 0.36$. Altogether, we simulated 24 data scenarios. For each scenario, we set $m = n = 100$ and simulate 20 replicates.

### 4.2.1 Performance on reconstruction error

We report the benchmark results in Figure 3. We utilize default setting of the benchmarking methods in our analysis. As for BABF, we assume the prior of $X, Y$ as Bernoulli distribution with $p_X = p_Y = 0.5$ and a flipping error of $p_f = 0.01$. The maximum iterations of $t_{all}, t_{BI}, t_{MF}$ are set at 20, 5 and 50.

For each method, we compare their performance using reconstruction_error, i.e, $|\hat{Z} - Z|$ as evaluation metric. Here, $|\cdot|$ represents the $L_1$ norm, and $\hat{Z}$ denotes the derived pattern matrix by each method. Lower reconstruction error indicates a better performance. It is anticipated that heuristic approaches like CG, MEBF, ASSO, PANDA would show varied performances respect to different data scenarios as different bias level would result in different impact on their underlying heuristic assumptions. Probabilistic method MP showed an overall stable performance but still struggles with high bias level. As expected, BABF achieves the most desirable performance with different bias levels, which highlights the importance to consider individual bias. Additionally, BABF revealed its robustness towards different data scenarios.

### 4.2.2 Evaluate inferred bias

We explore whether BABF could reliably recover the bias levels $\boldsymbol{\mu}, \boldsymbol{\nu}$. Here, we denote BABF inferred row- and column-wise bias as $\hat{\boldsymbol{\mu}}, \hat{\boldsymbol{\nu}}$. Since it is easy to find a scalar value $r$, s.t., $\boldsymbol{\mu}_i \cdot \boldsymbol{\nu}_j = r\hat{\boldsymbol{\mu}}_i \cdot \frac{1}{r}\hat{\boldsymbol{\nu}}_j$, we do not seek to directly compare the difference of values between $\boldsymbol{\mu}_i, \hat{\boldsymbol{\mu}}_i$, or $\boldsymbol{\nu}_j, \hat{\boldsymbol{\nu}}_j$, but instead analyze the correlation between the

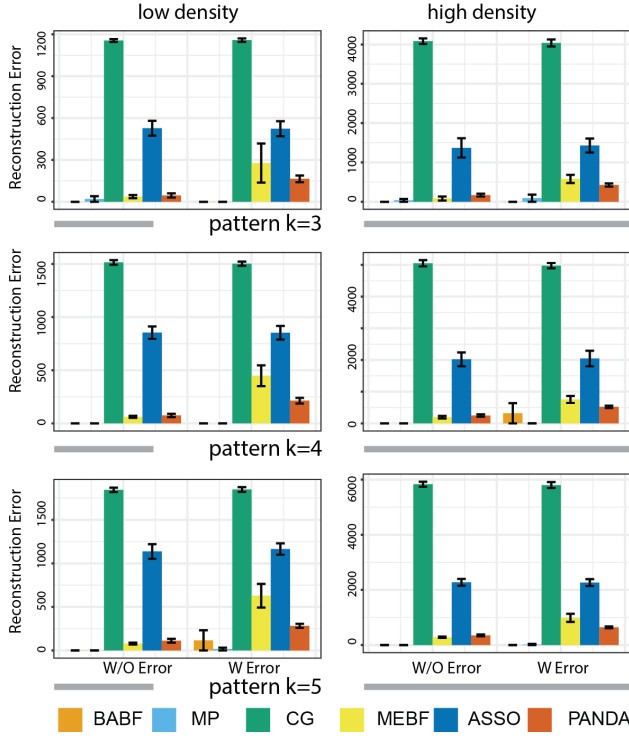

Figure 5: Performance comparison on simulated data without individual bias

inferred bias and true bias for every input matrix $A$, i.e., $corr(\boldsymbol{\mu}, \hat{\boldsymbol{\mu}}), corr(\boldsymbol{\nu}, \hat{\boldsymbol{\nu}})$. We report the correlation results across different data scenarios with pattern number ($k = 4$) in Figure 4. Every scenario has 20 replications. Figure 4A,B show the row- and column-wise bias across different data scenarios. In most cases, BABF inferred bias achieved over 0.8 correlation with ground truth. Even in the worst case, the correlation is as high as 0.4. To give a more intuitive idea, we reveal the inferred bias and true bias of the first input matrix from each scenario as an example in Figure 4C,D. The high correlation suggests desirable performance of BABF to infer the individual bias associated with the objects and the features.

### 4.2.3 Performance on data without bias

Next, we wish to test how BABF performs on data without bias ($\boldsymbol{\mu}_i = \boldsymbol{\nu}_j = 0, \forall i, j$). In other words, we would like to demonstrate that BABF works well in scenarios with or without bias. In Figure 3, we report the reconstruction error of the methods across 12 data scenarios all without background bias. In general, BABF and MP showed reliable performance. In some high density cases, BABF performs slightly worse than MP, but the difference is only marginal. Overall, BABF showed robust performance towards different data scenarios.

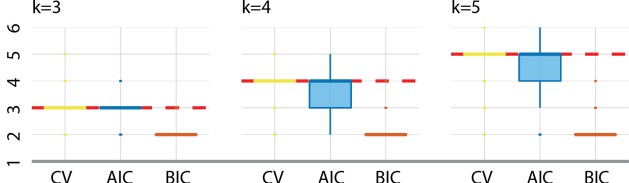

Figure 6: Model selection of pattern number $k$

Table 1: Reconstruction error on different stopping criteria

| $t_{BI}$ \ $t_{MF}$ | 10 | 25 | 50 | 100 |
|---|---|---|---|---|
| **5** | 56.1(242.0) | 2.2(5.5) | 2.2(5.5) | 33.0(.6) |
| **10** | 54.6(230.5) | 2.2(5.5) | 2.2(5.5) | 2.2(5.5) |
| **15** | 41.9(180.1) | 57.6(241.0) | 2.2(5.5) | 2.2(5.5) |

### 4.2.4 Selection of the pattern number

In our setting, the pattern number $k$ is the most important hyper-parameter that directly determines the number of variables in the factor graph. Under the probabilistic framework, we could utilize different statistical metrics to select the most optimal pattern numbers. Here we test three metrics, including cross validation accuracy (CV), Akaike information criterion (AIC) and Bayesian information criterion (BIC) for the model selection of $k$. For CV, we use 90% of the data for fitting and the rest 10% for testing [Kohavi et al., 1995]. For AIC and BIC, we utilize the formulation in Stoica and Selen [2004]. For all the methods, we evaluate the metrics on $k = \{2, ..., 6\}$ and select the best $k$ following their formulation. We tested above metrics across all 24 data scenarios and report the pattern number selections results in box plots (Figure 6). Here red dash marked the ground truth of $k$. Overall, CV showed consistently accurate selection of pattern number, with only marginal derivations for a small number of cases. AIC and BIC are impacted by the size of input data given a rather large number of variables in the model. Particularly, BIC tends to select a small $k$ for the model.

### 4.2.5 Testing on the stopping criteria

The optimization scheme of our bias aware BMF alternatively fits the bias and the pattern matrices: fitting pattern while giving bias corresponds to the algorithm component **prob_BMF**, and fitting bias while given pattern information corresponds to the algorithm component **Bias_infer**. At each iteration, we provide the option to set the maximum number of runs per step for **prob_BMF** and **Bias_infer** (corresponding to $t_{MF}$ and $t_{BI}$ in algorithm 1). In setting the correct $t_{MF}$ and $t_{BI}$, our goal is to find a rather *"optimal"* point that will not lead to premature overfitting of pattern or bias before the final convergence. We checked 12

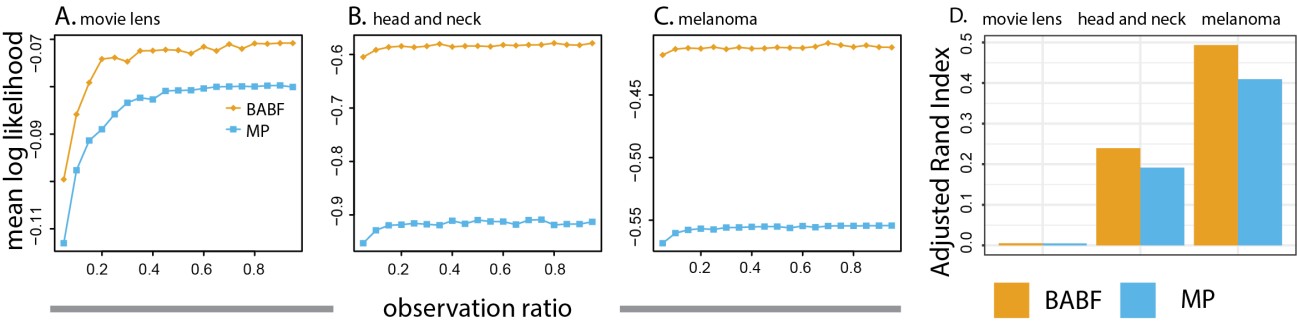

Figure 7: Goodness of fitting on the decomposition for real-world data

different combinations where $t_{MF} = 10, 25, 50, 100$ and $t_{BI} = 5, 10, 15$. For every combination, the maximum steps are set to 10000 to ensure convergence. We expect that $t_{MF}$ in general needs to be higher than $t_{BI}$ as the factor graph of **prob_BMF** is denser than **Bias_infer**. In table 1, we report the mean(standard derivation) reconstruction error of one scenario: high density, low bias, with noise and $k = 3$. As expected, it is not always higher the better for $t_{MF}$ or $t_{BI}$; instead, $t_{MF}$ and $t_{BI}$ need to be balanced to achieve small reconstruction error. Similar results can be seen across different scenarios. In practice we set $t_{MF} = 50$ and $t_{BI} = 10$ as default.

### 4.3 ANALYSIS ON REAL-WORLD DATA

We tested the performance of BABF on three real world datasets, movie lens data from Harper and Konstan [2015] and two biological gene expression datasets, head and neck cancer and melanoma single cell RNAseq (scRNA-seq) data from Puram et al. [2017], Tirosh et al. [2016]. The choice of the datasets as well as the pre-processing procedures follow previous works [Rukat et al., 2017, Wan et al., 2020c]. In movie lens data, we have 943 users that rated/not rated 1682 films. In head and neck, and melanoma data, we have 5902 cells that express/not express 7954 genes, and 4486 cells that express/not express 8210 genes, respectively. For each dataset, we first identify the number of patterns $k \in \{2, 20\}$ through cross validation, which yield 5 patterns in movie lens, 3 patterns in melanoma and 6 patterns in head and neck. BABF is then applied to retrieve $\hat{X}, \hat{Y}, \hat{\mu}$ and $\hat{\nu}$ following the specific pattern number for each dataset. We mainly focus on addressing two questions: 1. Would the consideration of individual bias benefit our interpretation of real world data? 2. Does the inferred bias carry any practical meaning?

#### 4.3.1 Data interpretation

Since the underlying true patterns of real world data is not accessible, instead of comparing decomposed pattern $\hat{Z}$ with input matrix $A$, where $A$ is constituted of not only the true

pattern matrix $Z$ but also likely noise matrix $E$ and bias matrix $B$, we use a likelihood metric. Specifically, we evaluate the goodness of model fitting as the overall likelihood, where a larger likelihood indicating a better fitting of the data. We compare the likelihood of BABF with the probabilistic BMF method MP. Similar to Ravanbakhsh et al. [2016], Rukat et al. [2017], we investigate the methods' performance by only keeping a certain percentage of the observations, called observation level, while masking the rest of the observations. At every observation level, we replicate the analysis for five times and report the mean log-likelihood value. We report the likelihood results in Figure 7A,B,C. On all three datasets, BABF showed higher overall likelihood compared with MP, which suggests that the individual bias assumption is more realistic for real-world data, that movie viewers or cells could be vastly different from each other even in the same pattern group, and such bias is independent with the latent pattern. This advocates the necessity to consider the individual bias in the BMF problem.

To further test the interpretability of the patterns, we examined how the patterns coincide with cell type labels in the two expression datasets, and the movie genres in the movie-lens dataset, using adjusted rand index, where a higher value corresponds to a greater similarity [Rand, 1971]. Figure 7D shows the peformance of BABF and MP on three datasets. Though both BABF and MP perform poorly on movie lens data, the decomposition from BABF showed higher similarity with given labels in both biological data, which partially revealed a better decomposition of BABF compared with MP.

#### 4.3.2 Practical meaning of inferred bias

The individual bias assumption allows BABF to outperform or have comparable performance with the existing BMF methods, whether such bias is present or not. Here, we want to understand whether the inferred bias could reflect certain practical meaning. Inspired by Wan et al. [2020a], in movie lens data, we want to explore the inferred bias on individual user with their taste on movie types. In our hypothesis, if a user only focus on certain genres of movie, then their

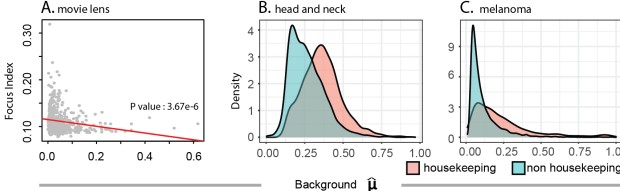

Figure 8: Interpretation analysis on inferred bias

## 6   ACKNOWLEDGMENT

The work is supported by NSF DBI IIBR 2047631, NSF IIS 2145314, American Cancer Society RSG-22-062-01-MM, NCI 5P30CA082709-22, NIA 1P30AG072976-01, and NIH NIGMS 1R01GM131399.

behavior could be majorly explained by pattern information $Z$, while with less effect from $B$. Here we design the *focus index* to quantify such effect. Specifically if a user watched $a$ movies in $c$ categories, i.e., $a = b_1 + ... + b_c$, the focus index of this user is calculated as $focus\_index := \sum_{i=1}^{c}(\frac{b_i}{a})^2$. As anticipated, inferred bias is negatively correlated with the focus index (Figure 8A, $corr = -0.19$, $p = 3.67e - 6$). This significant negative correlation revealed that the inferred bias partially revealed certain taste of the movie viewers.

In the case of gene expression, we focus on two groups of genes, housekeeping genes and non-housekeeping genes [Eisenberg and Levanon, 2013]. As the name revealed, housekeeping genes are to maintain the basic activities of the cells, that each cell, regardless of their cell types, will all express these genes. On the other hand, non-housekeeping genes will be the ones that reflect the cell-type specificity. For example, T cells will express T cell marker genes like CD3D,CD3E [Call et al., 2002, Wan et al., 2019]. Figure 8B,C are the density plot visualization of the inferred bias on housekeeping and non-housekeeping genes. As expected, housekeeping genes have a much bigger effect from bias as their expression behavior is not related with any patterns. On the other hand, since the non-housekeeping genes revealed the specificity of the cell, its behavior is largely covered by the latent pattern, such that we witness a small bias in $\hat{\mu}$ on both datasets.

## 5   CONCLUSION

In this paper, we propose a bias aware model, BABF, which is the first algorithm to derive Boolean matrix decomposition in the presence of individual object- and feature-wise bias. Compared with other methods, BABF is a highly efficient approach, which not only results in good approximation of the true binary pattern with low reconstruction error, but also infers individual bias with high consistency with ground truth. The bias inference from BABF could lead to interesting interpretations depending on different data scenarios.

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
