# OpenReview forum: "Bias Aware Probabilistic Boolean Matrix Factorization"
_auai.org/UAI/2022/Conference — UAI 2022 Poster_

### Official Review · Reviewer_Coqa · 2022-04-05

**Q2(1) Originality/Novelty:** 2
**Q2(2) Significance/Impact:** 2
**Q2(3) Correctness/Technical Quality:** 2
**Q2(6) Clarity Of Writing:** 2
**Q6 Overall Score:** 5
**Q8 Confidence In Your Score:** 5

**Q1 Summary And Contributions:**

This paper focuses on Boolean matrix factorization. It is motivated by the errors of observed data from true values are diverse in real-world scenarios while the existed works assume the noise model of data points is homeostatic. This paper proposes a bias-aware model BABF which takes an object- and feature-wise bias into consideration.

**Q2 Assessment Of The Paper:**

More detailed information regarding each of these aspects is given below:

**Q2(4) Quality Of Experiments (Optional):**

3: Good: The experimental evaluation is adequate, and the results convincingly support the main claims.

**Q2(5) Reproducibility:**

2: Fair: Key resources (e.g., proofs, code, data) are unavailable but key details (e.g., proof sketches, experimental setup) are sufficiently well-described for an expert to confidently reproduce the main results.

**Q3 Main Strengths:**

- This paper takes the object- and feature-wise bias into consideration whose motivation is clear.
- This paper makes adequate evaluations on both simulated data and real-world data across different settings. Partial experiment results verify the effectiveness of the proposed method.


**Q4 Main Weakness:**

- This paper is not well-organized and has some obvious syntax errors.
- Some figures in this paper are not easy to understand.
- The description of the method is not clear.
- The algorithm is a little complicated and the corresponding explanations are not followed by the main content, which makes it hard to understand.


**Q5 Detailed Comments To The Authors:**

- This paper uses many notations without explanations in “Introduction” which makes readers confused.
- Since Figure 1 has lots of subfigures, it would be better to use, e.g., (a) to substitute A to distinguish the index and input data.
- Some obvious syntax errors are found. For example, “for each scenarios...” on page 6. The authors should check the typos in this paper carefully.
- The logic of this paper is somewhat unreasonable. For example, the beginning of Section 4 summarizes the contents of this section in the first paragraph, then the following paragraph repeats part of the contents from the first paragraph which seems a bit redundant.
- More baselines need to be added for comparison on real-world data.
- One question: From what point of view can it be shown that the proposed method is efficient as you mentioned in the contributions.

**Q7 Justification For Your Score:**

The motivation and the logic of the whole paper are what I weigh most. This paper provides readers with a clear motivation and part of the experiment results verify the effectiveness of the proposed model. However, the logic is not well organized and some syntax errors are found. The description of the method is somewhat confusing to readers. The provided figures are not legible enough.

**Q9 Complying With Reviewing Instructions:**

1: Yes.

---

### Official Review · Reviewer_ZGPC · 2022-04-06

**Q2(1) Originality/Novelty:** 3
**Q2(2) Significance/Impact:** 2
**Q2(3) Correctness/Technical Quality:** 3
**Q2(6) Clarity Of Writing:** 2
**Q6 Overall Score:** 6
**Q8 Confidence In Your Score:** 4

**Q1 Summary And Contributions:**

This paper proposes a new probabilistic model for Boolean matrix factorization (BMF) including a bias, namely A = B + [(Z + E) mod 2) where Z is the low-rank BMF, and B is a rank-one bias. They also provide an algorithm to estimate the parameters of their model, dubbed BABF.

**Q2 Assessment Of The Paper:**

More detailed information regarding each of these aspects is given below:

**Q2(4) Quality Of Experiments (Optional):**

3: Good: The experimental evaluation is adequate, and the results convincingly support the main claims.

**Q2(5) Reproducibility:**

3: Good: Key resources (e.g., proofs, code, data) are available and key details (e.g., proofs, experimental setup) are sufficiently well-described for competent researchers to confidently reproduce the main results.

**Q3 Main Strengths:**

The new proposed model makes sense, as bias is a standard issue in low-rank models.

**Q4 Main Weakness:**

- The new model is not that surprising, since taking into account bias in data sets is a standard approach (it is for example widely used in recommender systems).
- The paper is not always clearly written. For example, it is very hard to follow Section 2.2 (the fact that equations are not properly punctuated does not help).
- There is not discussion in the paper about the convergence of the proposed algorithm. Also, there is not discussion on how the (many) initial factors are chosen and how this influences the algorithm results.
- The proposed algorithm iteratively updates the factors, X and Y, and the biases. For the factors, their update relies on previous works (Ravanbakhsh et al., 2016) so the algorithmic contribution is not that novel.

**Q5 Detailed Comments To The Authors:**

Other comments:
- I strongly recommend the authors to clarify Section 2.2 which is the basis for their model.
- NP-hardness of BMF is for the exact case (Stockmeyer), there are NP-hardness results in the literature for the approximate problem, even in the rank-one case. Morevoer, authors claim that NP-hardness remains for their biased variant, but they do not prove this. Although this makes sense and is likely to be true, they cannot claim such a result without proving it.


**Q7 Justification For Your Score:**

The model is intersting, and the numerical experiments are convincing. However, the paper is not well written (at parts), the novelty is limited, and the algorithm lacks some analysis (in particular, convergence, sensitivity to initialization --what initalization to use?).

**Q9 Complying With Reviewing Instructions:**

1: Yes.

---

### Official Review · Reviewer_9eyg · 2022-04-12

**Q2(1) Originality/Novelty:** 3
**Q2(2) Significance/Impact:** 2
**Q2(3) Correctness/Technical Quality:** 3
**Q2(6) Clarity Of Writing:** 4
**Q6 Overall Score:** 7
**Q8 Confidence In Your Score:** 3

**Q1 Summary And Contributions:**

The authors extend boolean matrix facrtorization for cases where individual rows and columns can have varying flipping noise rate, parameterized by real-valued numbers, presenting a reasonable learning algorithm and comprehensive empirical evaluation with good results.

**Q2 Assessment Of The Paper:**

More detailed information regarding each of these aspects is given below:

**Q2(4) Quality Of Experiments (Optional):**

4: Excellent: The experimental evaluation is comprehensive and the results are compelling.

**Q2(5) Reproducibility:**

3: Good: Key resources (e.g., proofs, code, data) are available and key details (e.g., proofs, experimental setup) are sufficiently well-described for competent researchers to confidently reproduce the main results.

**Q3 Main Strengths:**

Clearly defined problem formulation. High overall quality; good writing, comprehensive treatment, clearly described algorithms and good empirical experiments. I cannot easily think what could have been done better.

**Q4 Main Weakness:**

Slightly incremental as contribution, since the row/column biases are fairly straightforward to add into previous methods. The need for boolean MF could be motivated still a bit better and the relationship to binary MF that does not restrict the factors to be binary is unclear.

**Q5 Detailed Comments To The Authors:**

The work is in general of high quality. It is pleasant to read and the authors do a good job at describing the specific model and the learning algorithm. The combination of simulated and real experiments provides a broad perspective to the behavior of the algorithm and the empirical results are good. I really liked especially the simulated data experiments in Sec 4.2 and the way the results were presented graphically in Fig 4. The Supplement provides all technical details that are not trivial.

One major issue I have is that the paper is written solely from the perspective of boolean MF, directly assuming the problem formulation provided by the equation on the first page. The vast literature on approximating binary matrices with real-valued factors is not discussed at all and hence the motivation on why exactly we would want to constrain the solution this way remains unclear. I think it would be important to at least mention the alternative and discuss the main advantages and disadvantages of using binary factors. This is important especially because of your key contribution: Your goal is adding support for row/column biases that is easy for models that use real-valued X and Y, and you do that by introducing a separate rank-one factorization with real-valued parameters \mu and \nu. In effect, you are taking quite a big step towards models that directly use real-valued factors but still do not say anything about them. One could even argue that if you are willing to introduce some real-valued elements in your model then what is the value of constraining the factors to be binary? It complicates the learning problem and it is not clear whether all of the advantages of working with binary representations are retained.

**Q7 Justification For Your Score:**

Well conducted work with no obvious flaws in presentation or execution, but falls short of the best papers because the problem formulation feels a bit artificial and hence the impact is likely to be limited.

**Q9 Complying With Reviewing Instructions:**

1: Yes.

---

### Official Review · Reviewer_7Z33 · 2022-04-18

**Q2(1) Originality/Novelty:** 2
**Q2(2) Significance/Impact:** 2
**Q2(3) Correctness/Technical Quality:** 3
**Q2(6) Clarity Of Writing:** 3
**Q6 Overall Score:** 5
**Q8 Confidence In Your Score:** 3

**Q1 Summary And Contributions:**

This paper introduces a method for Boolean
matrix factorization that takes object and
feature bias into account.


**Q2 Assessment Of The Paper:**

More detailed information regarding each of these aspects is given below:

**Q2(4) Quality Of Experiments (Optional):**

2: Fair: The experimental evaluation is weak: important baselines are missing, or the results do not adequately support the main claims.

**Q2(5) Reproducibility:**

2: Fair: Key resources (e.g., proofs, code, data) are unavailable but key details (e.g., proof sketches, experimental setup) are sufficiently well-described for an expert to confidently reproduce the main results.

**Q3 Main Strengths:**

Cool and plausible variation on BMF that nobody has taken a crack at before - those
aren't easy to find.


**Q4 Main Weakness:**

I would feel a lot more positive about this paper
if I could think of more applications from which I
could sample non-simulated data with which to
test it.  I don't doubt the existence of
bias, but (1) I would think it would appear
mainly in matrices with floating-point numbers
and (2) I'm not sure how often that basis would
be attributable to a heteroscedastic error
distribution in particular.

The authors do include an evaluation on some
real-world data.  Is there a test like Levene's
that you could run to determine how much of
the error is attributable to heteroscedasticity?

Don't know how to get hold of any of this data
(hence my reproducibility score).

The message passing appendix seems essential
for understanding and reproducing this result, and so
perhaps an 8-page conference paper is not the right
format for this work.


**Q5 Detailed Comments To The Authors:**

p. 1: treat all data points equally distributed -> ... as equally distributed

p. 2: assumption on homoscedastic -> of

p. 3: often yield -> yields
  such strategy -> a strategy

p. 4: our observations A is generated -> are


**Q7 Justification For Your Score:**

See weaknesses above.  I am willing to adjust my score based upon the
response tendered to each of these concerns.


**Q9 Complying With Reviewing Instructions:**

1: Yes.

---

### Decision · Program_Chairs · 2022-05-15

**Decision:**

Accept (Poster)

**Comment:**

Meta Review: In this paper, the authors propose a new probabilistic model for Boolean matrix factorization (BMF) including a bias. All reviewers consider that this paper is well written, and the problem definition is clearly defined. And the authors also address the concerns of reviewers after the rebuttal.